# Exome sequencing and analysis of 44,028 British South Asians enriched for high autozygosity

Genes & Health (G&H) is a biomedical study of adult British Pakistani and Bangladeshi research volunteers enriched for autozygosity. Here we performed whole-exome sequencing in 44,028 G&H participants, establishing a large publicly available South Asian exome resource linked to longitudinal electronic health records. We performed exome-wide association analyses for 645 electronic health record-derived traits under additive and recessive models, and meta-analyses of 33 cardiometabolic traits with UK Biobank, finding more than 100 novel gene–phenotype associations. We identified 2,991 genes with rare biallelic predicted loss-of-function ('knockout') genotypes, 546 of which had not been previously reported. We show that drugs targeting genes with knockouts in adults are associated with a 2.2-fold higher likelihood of progressing beyond phase 1 clinical trials. We further illustrate how phenotypic profiles associated with knockout genotypes can enhance efficacy and safety assessment of drug targets and aid in the interpretation of variants with ambiguous clinical significance in autosomal recessive disease genes.

Major advances in our understanding of human diseases have been achieved through genotyping, sequencing and analyses of large population- or hospital-based cohorts[1–5]. However, there remains a critical underrepresentation of non-European ancestral groups in genetic datasets[6,7]. Addressing this imbalance is essential not only for ensuring equity in genetic research but also for maximizing discoveries by harnessing distinct variant spectra present in diverse ancestry groups[8–11].

Genes & Health (G&H) is a population-based cohort study of British Bangladeshi and Pakistani adults, aiming to improve the health outcomes of these communities through genetic research. G&H benefits from comprehensive lifelong healthcare data from the UK National Health Service (NHS), which allows systematic genetic association analyses of disease diagnoses and clinical traits[12,13] and detailed medical record reviews in specific carriers of interest[14,15].

A distinctive characteristic of this cohort is its high degree of autozygosity and, hence, enrichment of rare homozygous genotypes[16,17]. Particularly informative are homozygous carriers of loss-of-function variants, often called human knockouts. Studies from this cohort and others have demonstrated the valuable insights that human knockouts can provide into human biology, disease mechanisms and drug development[14,16,18–20].

Here, we present analyses of whole-exome sequences of 44,028 participants in G&H, representing a large South Asian exome resource with linked electronic medical data. We share key findings from rare variant association analyses, using both additive and recessive models, alongside meta-analyses with UK Biobank (UKB) that uncovered numerous novel gene–phenotype associations. We further highlight clinical and drug development insights gained from the human knockouts identified in this cohort.

## Results

### Protein coding variation and population structure in 44,028 South Asian exome sequences

We identified a total of 4,723,926 variants (4,458,984 single-nucleotide variants (SNVs) and 264,942 insertions or deletions (INDELs)) from 44,028 G&H exome sequences after stringent quality control (Supplementary Methods). Across 17,545 transcripts from the Matched Annotation from the NCBI and EMBL-EBI (MANE) project[21], we found 122,690

✉e-mail: hyein.kim@pfizer.com; hcm@sanger.ac.uk; d.vanheel@qmul.ac.uk

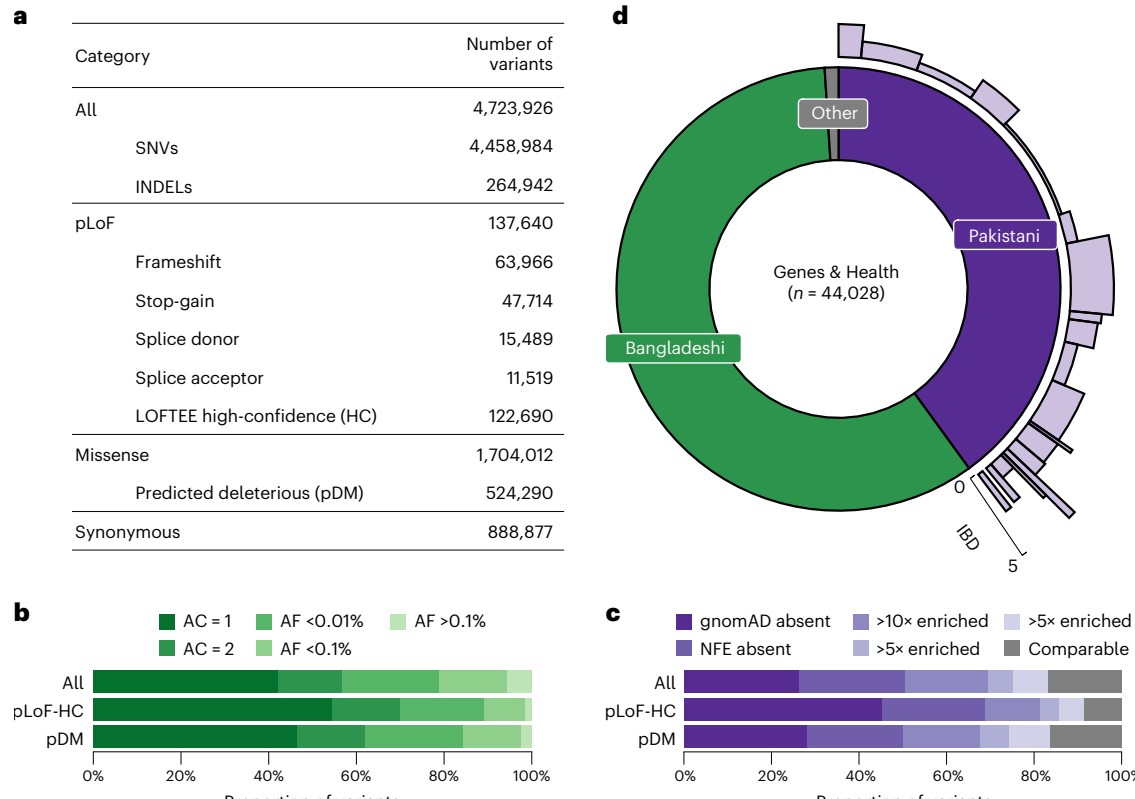

**Fig. 1 | Protein coding variation and population structure in 44,028 G&H exomes. a**, Number of all variants, including SNVs and INDELs, broken down by functional category. **b**, AF spectrum of all, pLoF-HC and pDM variants. **c**, Proportions of variants completely absent in gnomAD, absent in the NFE subset of gnomAD, or with 10×, 5× or 2× greater AF compared with gnomAD-NFE.

The gray portion includes variants in both G&H and gnomAD with comparable AF. **d**, South Asian ancestry breakdown in G&H and subpopulation structure among British Pakistanis. For the British Pakistani subpopulations in light purple, the width indicates the relative proportion of clusters, and the height indicates the IBD score in the clusters.

predicted loss-of-function (pLoF) variants that are high-confidence (pLoF-HC) by LOFTEE[22], and 1,704,012 missense variants, of which 524,290 are predicted damaging missense (pDM) (Combined Annotation-Dependent Depletion (CADD) score >20, Polymorphism Phenotyping v2 (Polyphen2) score >0.445 and predicted deleterious by Sorting Intolerant From Tolerant (SIFT)) (Fig. 1a). As expected, pLoF and pDM variants were heavily enriched among singleton and ultrarare variants (Fig. 1b). We compared the allele frequency (AF) of the variants in G&H with those in the Genome Aggregation Database[22] (gnomAD, v4.1), which catalogs variants from 807,162 genomes and exomes of diverse ancestry, including 45,546 of South Asian ancestry. Among all the variants in G&H, 26.2% are absent from gnomAD, a further 24.4% are in gnomAD but not in the non-Finnish European (NFE) subset, and a further 18.8% have >10-fold higher AF in G&H compared with gnomAD-NFE (Fig. 1c). Compared with gnomAD-NFE, 325,276 variants were significantly enriched in 17,172 unrelated individuals of G&H (Methods) (Fisher's exact test $P < 4.93 \times 10^{-8}$ with Bonferroni correction).

G&H consists of British residents of self-identified Pakistani (40%) and Bangladeshi (59%) ancestry (Fig. 1d). The cohort has a high rate of self-reported parental relatedness (22.8% related; 15.1% first cousins) and autozygosity. Compared with the European-ancestry subset of UKB, the genetically determined rate of consanguinity (offspring of second cousins or closer) was higher in G&H (33% compared with 2%), and so was the average fraction of the genome in runs of homozygosity (1.8% compared with 0.4%)[17]. Principal component analysis demonstrates that there is considerable population structure among the British Pakistanis but much less among the British Bangladeshis[23] (Supplementary Fig. 1). The population structure of British Pakistanis is strongly influenced by the biraderi social stratification system and is characterized by

extensive identity-by-descent (IBD) sharing due to founder effects[24]. Using an IBD-based clustering method[24] (Methods), we identified 21 clusters among 8,109 unrelated British Pakistani individuals (Fig. 1d, Supplementary Figs. 1 and 2, and Supplementary Table 1) representing putative subgroups. Several clusters have particularly extensive IBD sharing (Fig. 1d and Supplementary Fig. 2). We identified 15,200 variants that were significantly enriched in specific clusters compared with all the others combined, which may have resulted from founder events or possibly positive selection, described further in Supplementary Note 1 (Supplementary Figs. 2 and 3, and Supplementary Tables 2 and 3).

There were 8,450 variants across 2,855 genes that were curated as pathogenic or likely pathogenic (PLP) in ClinVar[25] and had at least one heterozygous or homozygous genotype in G&H exomes (Supplementary Note 2, Supplementary Fig. 4a and Supplementary Table 4). Among the 81 clinically actionable genes defined by the American College of Medical Genetics and Genomics[26] (ACMG SF v3.2), we found 1,012 individuals heterozygous for PLP variants in autosomal dominant genes and 7 individuals homozygous for PLP variants in autosomal recessive (AR) genes. Including pLoF variants that are previously unannotated by ClinVar in genes with a known loss-of-function mechanism further increased these numbers (Supplementary Note 2). Compared with a size-matched subset of European-ancestry exomes from UKB (UKB-EUR), a smaller portion of pLoF and pDM variants in G&H were already present in ClinVar, and a smaller portion of those in ClinVar were annotated as pathogenic or of uncertain significance (Supplementary Note 2, Supplementary Fig. 4b,c and Supplementary Table 5). These presumably reflect the relative paucity of patients of South Asian ancestry who have undergone clinical sequencing, as well as potential geographical differences in the practice of reporting variants to ClinVar.

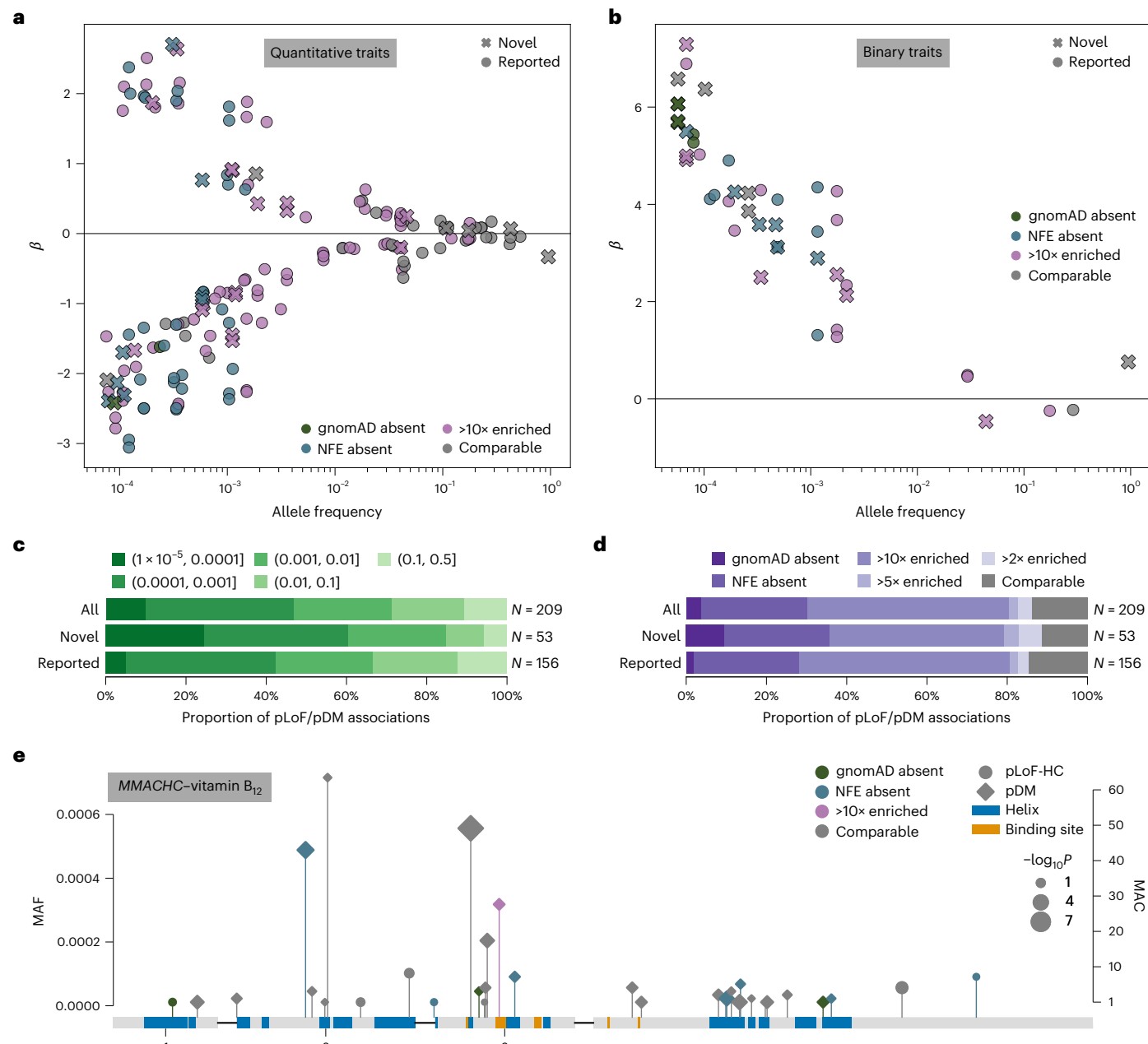

**Fig. 2 | Rare variant association analyses. a,b,** Single-variant MAF versus effect size ($\beta$) for significant quantitative (**a**) and binary (**b**) trait associations. Circle markers, pLoF variants; diamond markers, pDM variants; dark marker outlines, novel gene–phenotype association; purple marker colors, G&H MAF enrichment. **c,** Proportion of pLoF and pDM single-variant associations annotated as novel or previously reported, stratified by MAF bin. **d,** Proportion of pLoF and pDM single-variant associations annotated as novel or previously reported, stratified by AF ratio between G&H and gnomAD-NFE. **e,** pLoF-HC and pDM variants in *MMACHC* with MAF <0.001 that were included in the gene-based test with significant burden association with vitamin $B_{12}$ levels. Purple marker colors, G&H MAF enrichment; circle markers, pLoF variants; diamond markers, pDM variants. Blue and orange boxes indicate UniProt alpha helices and binding sites, respectively. Marker size indicates $-\log_{10}P$ values from the association tests for individual variants; this is on a continuous scale, with just three representative marker sizes shown in the legend.

## Rare variant association analyses
### Exome-wide association analyses across 645 EHR-derived traits.
We performed exome-wide association analyses using REGENIE[27] for 54 quantitative traits extracted from routine clinical and laboratory measurements and 591 binary traits derived from diagnosis and clinical procedure codes (Supplementary Table 6 and Supplementary Methods). Given that many phenotypes, variants and gene masks are correlated, we performed permutations at the second step of REGENIE to determine the *P*-value threshold corresponding to a false discovery rate (FDR) of 5% (Supplementary Methods) rather than applying an overly conservative Bonferroni correction ($P < 0.05/1,122,594,120$ tests $= 4.45 \times 10^{-11}$) (see Supplementary Note 3 for comparison). We permuted the genotypes instead of the phenotypes to control for the fine-scale population structure and relatedness in the samples. There was minimal genomic inflation in the summary statistics, indicating that REGENIE was adequately controlling for these potential confounders (Supplementary Note 4 and Supplementary Table 7).

In total, we found 2,982 single-variant and 907 gene-based associations (FDR <5%; Supplementary Tables 8 and 9). Among the significant single-variant associations, 265 involved variants that are

significantly enriched in specific British Pakistani subpopulations, and 218 out of these 265 (82%) involved human leukocyte antigen (HLA) variants (Supplementary Note 1). We removed associations involving HLA variants/genes, synonymous gene masks and associations that were not significant after conditioning on nearby genome-wide association study (GWAS) associations, resulting in the final set of 1,191 single-variant and 722 gene-based associations (Extended Data Fig. 1). These included many known gene–phenotype relationships, and in many cases, variants that were either private or enriched in G&H further expanded the allelic series (Supplementary Note 4 and Supplementary Fig. 5). We attempted replication of the significant associations from G&H in an independent set of 11,863 exomes from G&H (G&H 12k) and in the 430,998 European-ancestry exomes of UKB[28]. For single-variant and gene-based associations, respectively, we replicated 86.3% and 85.8% at nominal $P < 0.05$ and 40.2% and 66.7% at Bonferroni-corrected $P < 0.05$ in either replication dataset (Extended Data Fig. 2, Supplementary Table 10 and Methods).

We assessed whether gene–phenotype pairs with significant associations in G&H have previously been implicated by GWAS or rare variant association studies (Methods). The single-variant and gene-based associations involved 169 unique gene–phenotype pairs for quantitative traits and 40 for binary traits (Supplementary Fig. 6). Among these 209 gene–phenotype pairs, 66 (32%) did not have prior genetic associations, and we term these 'novel' (Supplementary Table 11). Variants that are absent in gnomAD or enriched in G&H tended to be rarer and have larger effects on the phenotypes, driving many novel gene–phenotype relationships (Fig. 2a,b). Specifically, among pLoF/pDM single-variant associations, variants in lower minor AF (MAF) bins were more likely to implicate novel gene–phenotype pairs (Cochran–Armitage test, $z = 3.37$, $P = 7.50 \times 10^{-4}$; Fig. 2c), as were variants absent from gnomAD (Fisher's exact test, odds ratio (OR) 5.31, $P = 0.026$; Fig. 2d).

We highlight two associations that are driven by variants private to or enriched in G&H (for further details, see Supplementary Note 4). First, we identified novel associations between three rare (MAF <0.2%) pDM variants in *ABCB6* and increased serum potassium levels. *ABCB6* encodes an erythrocyte membrane ABC transporter, and rare *ABCB6* missense variants have been reported to cause autosomal dominant pseudohyperkalemia, a temperature-dependent passive leak of red blood cell potassium into plasma[29]. One variant, p.Arg375Trp (chr2-219216028-G-A), is highly enriched in G&H (MAF $3.4 \times 10^{-4}$) compared with gnomAD-NFE (MAF $8.5 \times 10^{-7}$) and coincides with a previously reported pedigree for pseudohyperkalemia from East London[30]. Among these three pDM associations, one replicated at Bonferroni-corrected $P < 0.05$, another replicated at nominal $P < 0.05$, and the last was too rare to test in the G&H 12k replication cohort. Another example is the association between the pLoF/pDM burden of *MMACHC* and vitamin $B_{12}$ levels ($P = 1.61 \times 10^{-11}$, $\beta = 0.49$) with nominal replication in G&H 12k ($P = 0.004$). *MMACHC* encodes a vitamin $B_{12}$ transporter, and rare *MMACHC* mutations cause a vitamin $B_{12}$ disorder, methylmalonic aciduria and homocystinuria, cblC type[31]. A noncoding variant 12 kb upstream has been linked to homocysteine levels[32], and a previous study reported a single *MMACHC* missense variant associated with $B_{12}$ levels[33]; here, we report multiple coding variants in *MMACHC* associated with vitamin $B_{12}$ levels, illustrating a compelling allelic series. Among the 33 variants in the burden, 3 are absent in gnomAD, 8 are absent from gnomAD-NFE, and 1 is 187-fold more frequent in G&H (MAF $3.18 \times 10^{-4}$) compared with gnomAD-NFE (MAF $1.70 \times 10^{-6}$) (Fig. 2e).

**Meta-analyses of 33 cardiometabolic traits with UKB.** Cross-ancestry meta-analysis may benefit from increased allelic diversity within a gene and number of carriers leading to improved statistical power for discovery. Because British Bangladeshi and Pakistani communities have the highest prevalence of cardiometabolic diseases in the UK[34], we performed meta-analyses for select cardiometabolic traits between 44,028 G&H participants and 409,499 European-ancestry UKB participants (Methods).

From single-variant analyses, we identified 1,738 significant associations with consistent effect direction between G&H and UKB ($P < 3.3 \times 10^{-8}$ for binary traits, $P < 7.5 \times 10^{-9}$ for quantitative traits), 399 of which were not identified in either cohort alone (Supplementary Table 12). There were 146 associations for pLoF or pDM variants, 24 of which only became significant in the meta-analysis (Fig. 3a). From gene-based analyses, we identified 577 significant associations ($P < 3.5 \times 10^{-8}$ for binary traits, $P < 4.5 \times 10^{-7}$ for quantitative traits), comprising 139 unique gene–phenotype pairs (Supplementary Table 13). Among these, 21 rose to significance only in the meta-analysis (Fig. 3b). In both single-variant and gene-based analyses, the benefit of meta-analysis was particularly prominent for binary traits, with 41% and 50% of the significant associations, respectively, being only identified in the meta-analysis.

To explore the degree of power gain G&H contributed to the meta-analysis, we compared the $P$ values from the meta-analysis with those from UKB alone (Fig. 3c,d and Extended Data Fig. 3). The greatest power gain was seen for binary traits related to circulatory system diseases in both single-variant and gene-based results. In fact, the associations for these circulatory system diseases had stronger $P$ values in G&H compared with UKB, despite the nearly 10-fold difference in sample size between the two cohorts (Supplementary Fig. 7) and lower case prevalence in G&H compared with UKB (Supplementary Table 6). Both the frequency and effect size of the variants were greater in G&H compared with UKB (Supplementary Fig. 7), probably contributing to the stronger $P$ values. This suggests that power gain in cross-ancestry meta-analysis can be influenced not only by the difference in variant frequency spectrum but also by the difference in variant penetrance between ancestry groups.

Among the 21 and 24 gene–phenotype pairs newly implicated by gene-based and single-variant meta-analyses, 14 and 13 were novel, respectively. One notable example is the association between the pLoF/pDM burden of *LMNA* and atrial fibrillation and flutter (OR 1.52, $P = 9.5 \times 10^{-9}$) driven by rich allelic diversity in G&H and UKB (Fig. 3d). Interestingly, the effect size of the burden was much greater in G&H than in UKB (heterogeneity $P = 5.51 \times 10^{-4}$) (Fig. 3d). A smaller candidate gene study has suggested a potential link between *LMNA* and lone atrial fibrillation[35]; here, we report evidence from unbiased genetic association analyses. This association was minimally affected by the removal of individuals with dilated cardiomyopathy (I42), a rare condition previously linked to *LMNA*, from the analysis. Another example is the association between *ADCY6* singleton pLoF-HC burden and intracerebral hemorrhage (OR 326, $P = 3.7 \times 10^{-9}$). *ADCY6* encodes a member of the adenylyl cyclase protein family and plays an important role in maintaining a homeostatic contractile state of smooth muscle cells in the vessel wall and in regulating blood pressure[36], which may explain its association with intracerebral hemorrhage. Biallelic mutations (mostly missense and one splice donor) in this gene have been reported to cause a lethal congenital contractures syndrome[37] (Online Mendelian Inheritance in Man (OMIM) 616287). Lastly, a rare stop-gain variant (chr14-22773945-G-A, p.Arg473Ter) in *SLC7A7* was associated with atherosclerosis (OR 74.28, $P = 4.7 \times 10^{-9}$). This variant is pathogenic in ClinVar for lysinuric protein intolerance, a rare AR genetic disorder caused by impaired metabolism of lysine. Dysregulation in lysine metabolism has been linked to cardiometabolic pathophysiology[38,39], which may influence the risk of atherosclerosis.

**Recessive burden analyses with biallelic genotypes.** High autozygosity in G&H can provide greater statistical power for recessive association analyses, which have been relatively less explored[40–43]. There were 13,821 and 110,194 homozygous genotypes for pLoF and pDM variants with MAF <5%, respectively (Supplementary Table 14). We performed statistical phasing (Supplementary Methods and Supplementary Fig. 8a) to identify compound heterozygous genotypes[43], further increasing the number of biallelic pLoF and pDM genotypes by 45% (Extended Data Fig. 4a

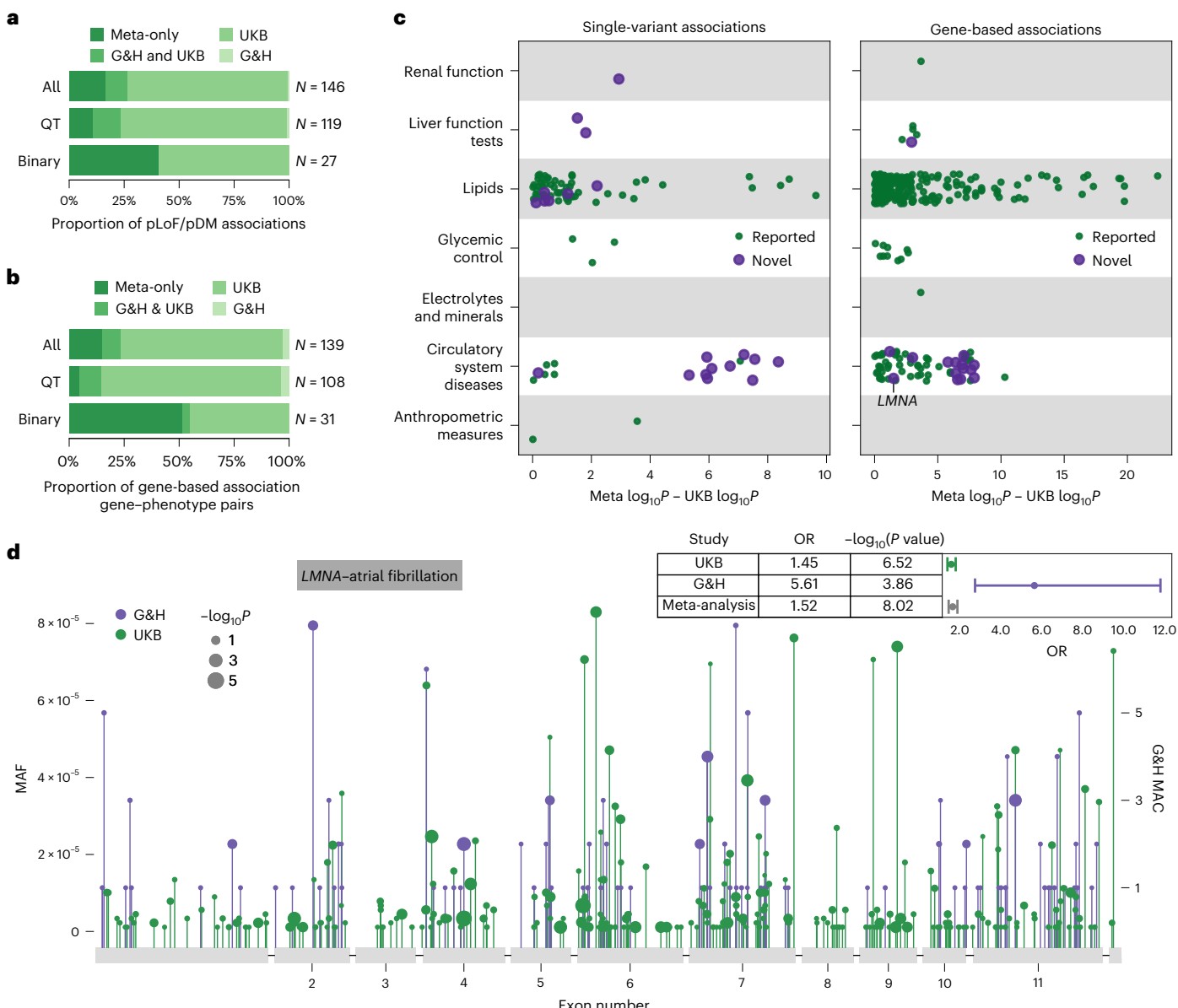

**Fig. 3 | Meta-analysis and recessive burden analyses. a,b,** Proportion of pLoF and pDM single-variant associations (**a**) and proportion of gene–phenotype pairs with gene-based associations (**b**) stratified by the significance status in respective studies and by quantitative (QT) and binary traits: only in the meta-analysis; in both G&H and UKB; in UKB only; or in G&H only. **c,** Difference in −log₁₀ association *P* values between the meta-analysis and the UKB-only analysis for significant single-variant pLoF/pDM associations (left) and gene-based associations (right) from the meta-analysis. Marker color indicates that the

gene–phenotype association is annotated as novel. **d,** Lollipop plot displaying the location of all pLoF and missense variants with MAF <0.0001 in *LMNA* in G&H (in purple) and UKB (in green). The *y* axis represents the MAF or MAC. The marker size corresponds to −log₁₀*P* values from the association tests for individual variants; this is on a continuous scale, with just three representative marker sizes shown in the legend. The forest plot in the top right corner shows the ORs and *P* values from G&H, UKB and meta-analysis for the gene-based association between *LMNA* and atrial fibrillation.

and Supplementary Table 14). Individual genotypes were then collapsed into three gene burdens, one with biallelic pLoF genotypes alone, another with biallelic pLoF and pDM genotypes, and the third with biallelic synonymous genotypes as negative control. We estimated that, on average across all genes that had at least one biallelic pLoF or pDM genotype, the high autozygosity resulted in 2.44-fold increase in the total number of biallelic genotypes compared with what would be expected under Hardy–Weinberg equilibrium. In Supplementary Note 5 and Supplementary Fig. 9, we illustrate the increase in the statistical power for recessive effects that we gain from autozygosity given a range of parameters. Using REGENIE[27], we performed a recessive gene-based test with gene burdens with at least 4 biallelic carriers for 54 quantitative and 439 binary traits (Supplementary Table 6).

We found 13 significant associations under the recessive model ($P < 2.89 \times 10^{-7}$, FDR ~7.14%; Supplementary Fig. 8b, Supplementary Table 15 and Supplementary Methods), many of which had stronger *P* values and effect sizes under the recessive model than under the additive model (Supplementary Fig. 10). To identify associations with non-additive effects, we tested for dominance deviation by jointly modeling the additive and dominant effects (Methods). Three associations had significant dominance deviation ($P_{domdev} < 0.05/13 = 0.0038$) with a clear recessive phenotypic pattern (Extended Data Fig. 4b,c) and had no prior associations linking the gene to the phenotype. First is an association of *NLRP10* with viral pneumonia ($P_{rec} = 6.11 \times 10^{-8}$; $P_{domdev}$ = 0.0028) (Extended Data Fig. 4b) along with a suggestive association with 'viral agents causing diseases classified elsewhere' ($P_{rec} = 4.01 \times 10^{-6}$;

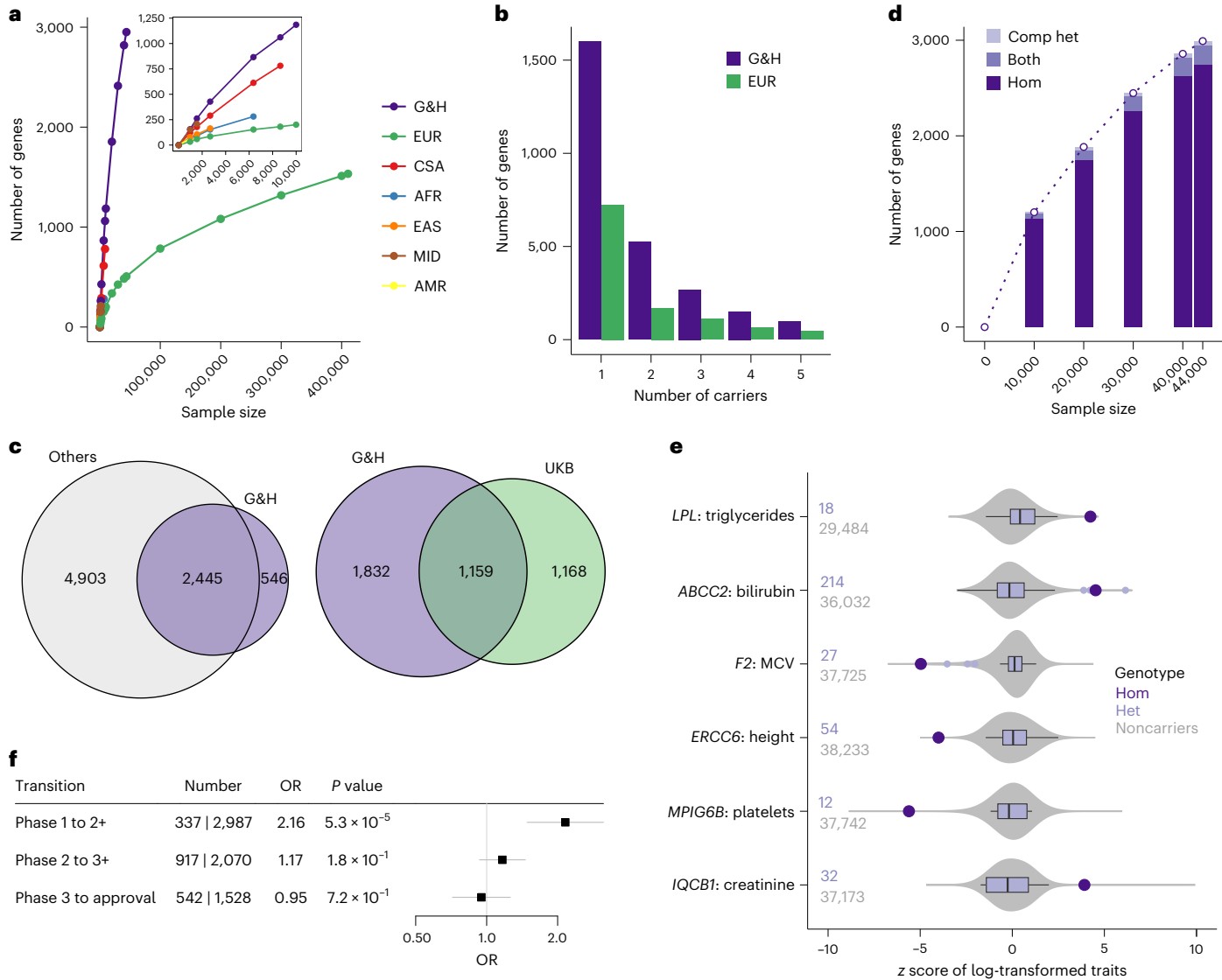

**Fig. 4 | Identification of genes with human knockouts and insights into drug development and clinical variant interpretation. a,** Accrual of genes with at least one homozygous pLoF genotype in G&H and UKB exomes stratified by ancestry groups. Inset: magnified data points for smaller sample sizes. EUR, European; CSA, Central/South Asian; AFR, African; EAS, East Asian; MID, Middle Eastern; AMR, admixed American ancestry. **b,** Number of genes with up to 5 pLoF homozygous genotypes in 44,028 G&H exomes and 469,814 European-ancestry exomes from UKB. **c,** Overlap of genes with human knockouts in G&H exomes compared with those in four other genomic datasets (left) or with those in UKB exomes (right). **d,** Accrual of genes with at least one human knockout (either homozygous or compound heterozygous (comp het) genotype) at increasing sample sizes of G&H exomes. **e,** Disease-relevant quantitative phenotypes in the homozygous carriers of pLoF variants in AR disease genes. Large points in dark purple indicate the lifetime median values of the homozygous ('Hom') carriers. Box plots in light purple show the distribution of values among the heterozygous ('Het') carriers. The center lines indicate the median values. The edges of the box indicate the first and third quartiles. The whiskers extend to the most extreme data points within 1.5 times the interquartile range. The points indicate the data points that fall outside the range of the whiskers. Gray violin plots show the distribution of the values among the noncarriers. The number of heterozygous carriers and noncarriers are indicated to the left of the plots. **f,** Enrichment analysis results for antagonistic drugs with human knockouts per clinical trial phase using logistic regression test. The centers and the error bars indicate the ORs and the 95% confidence intervals, respectively.

$P_{domdev}$ = 0.0028), consistent with a key role of *NLRP10* in the inflammasome pathway[44]. Next, *HSD17B14*, a gene involved in steroid hormone metabolism[45] with possible indirect impact on bone health, was associated with disorders of bone density ($P_{rec}$ = 7.12 × 10⁻⁸; $P_{domdev}$ = 2.0 × 10⁻⁵) (Extended Data Fig. 4b). Lastly, we found an association between *NCAPD2* and vitamin B₁₂ ($P_{rec}$ = 2.60 × 10⁻⁷; $P_{domdev}$ = 2.48 × 10⁻⁴) (Extended Data Fig. 4c). *NCAPD2* encodes a condensin I complex subunit essential for chromosome condensation[46], and additional studies are needed to establish its relevance to B₁₂ metabolism. More details on these and additional suggestive associations ($P_{rec}$ < 5.0 × 10⁻⁶) are provided in Supplementary Note 6 and Supplementary Table 15. Overall, these results suggest that increasing the sample size of biallelic carriers, especially

those of rare variants, for recessive association testing may yield further novel findings that may be missed by additive association testing.

## Insights from human knockouts

**Discovery of 2,991 genes with putative human knockouts.** G&H exomes, as expected, showed higher accrual rate of genes with one or more homozygous pLoF-HC genotypes compared with the ancestry groups in UKB that originate from populations with low consanguinity (Fig. 4a,b). In the 44,028 G&H exomes, we identified a total of 2,991 genes with biallelic pLoF genotypes, referred to as putative human knockouts (Supplementary Table 16): 2,951 genes with 8,144 homozygous genotypes and 249 genes with 473 compound heterozygous genotypes. Genes with

biallelic loss in G&H were depleted among the genes that are essential in cell culture, knockout lethal in mice and implicated in AR diseases (Supplementary Note 8, Supplementary Fig. 11 and Supplementary Table 17).

We compared the list of genes with human knockouts in G&H with those found in other genomic datasets, although some of these studies may not include phenotype information as detailed as that available in G&H. Aggregating across five datasets (gnomAD[22] v4 exomes, which includes UKB, RGC-ME[47], deCODE[18], PROMIS[19] and Born in Bradford[16]), there were 7,348 genes with human knockouts in approximately over 1.4 million individuals. Despite the much smaller sample size, 546 genes were found to have human knockouts only in G&H (Fig. 4c, left). We also found that 1,669 genes have human knockouts only in G&H compared with 2,327 genes with human knockouts in the 430,771 exomes of UKB[28], a cohort with broadly accessible phenotypic information (Fig. 4c, right).

In G&H, unlike the ancestry groups with low consanguinity in UKB, the number of genes with human knockouts is growing close to linearly at the current sample size (Fig. 4a,d), suggesting that sequencing of additional individuals will identify more genes with biallelic loss. Because the average level of autozygosity is higher among British Pakistanis compared with British Bangladeshis[17], sequencing British Pakistanis is generally expected to find more homozygous pLoF genotypes. However, there is slightly reduced genetic diversity in Pakistanis compared with Bangladeshis, resulting from the historic bottleneck events mentioned above[24]. We found that, when conditioned on the level of autozygosity and sample size, sequencing British Bangladeshis can maximize the number of unique pLoF variants (and genes) with at least one homozygous genotype, while sequencing British Pakistanis can maximize the number of pLoF variants with more than one homozygous genotype (Supplementary Note 7, Supplementary Fig. 12 and Supplementary Methods).

**Clinical utility of human knockouts in variant interpretation for AR disease genes.** The high autozygosity and rich phenotypic information in G&H can facilitate the assessment of the clinical impact of variants in AR disease genes. Although pLoF variants in disease genes with a known loss-of-function mechanism are often automatically classified as likely pathogenic, computational predictions may be erroneous[22]. We found 368 Mendelian disease genes with AR inheritance[48] that have individuals homozygous for pLoF variants in G&H (Supplementary Fig. 4d and Supplementary Table 18). Among these pLoF variants, 63% are unreported in ClinVar and 12% have uncertain significance or conflicting interpretations (VUS/CI). We inspected the health records of the homozygous carriers of these variants and identified several examples where they provided supporting evidence that the variants are indeed pathogenic (Supplementary Table 19). One example is a stop-gain variant in *LPL* (chr8-19955849-C-T, p.Gln262Ter) with conflicting interpretations in ClinVar. Loss of this gene causes lipoprotein lipase deficiency (OMIM 238600) characterized by highly elevated serum triglycerides and ectopic lipid deposition. One homozygous carrier of this variant had significantly elevated serum triglyceride levels from their early 30s (lifetime median of 15.7 compared with 1.5 mM among the rest of the cohort; $z$-test $P = 2.22 \times 10^{-5}$), lowered high-density lipoprotein cholesterol levels (0.4 versus 1.2 mM; $z$-test $P = 3.47 \times 10^{-6}$), diagnostic codes for E78 (disorders of lipoprotein metabolism) and other related complications including K85 (acute pancreatitis), type 2 diabetes, and steatohepatitis, and prescriptions of Omacor tablets (omega-3-acid ethyl esters used for hypertriglyceridemia). Another example is a frameshift variant in *ABCC2* (chr10-99818879-CCT-C, p.Leu788ValfsTer13) with conflicting interpretations in ClinVar. Loss of this gene is implicated in Dubin–Johnson syndrome (OMIM 237500) with clinical manifestation of chronic cholestatic jaundice. One homozygous carrier of this variant had persistently raised bilirubin from their early 30s (56 versus 7 µM; $z$-test $P = 6.06 \times 10^{-6}$) and diagnosis codes for E80 (disorders of porphyrin and bilirubin metabolism). This phenotypic profile is comparable to that observed in another homozygous carrier of a likely pathogenic splice donor variant (chr10-99792360-G-A), including consistently elevated serum bilirubin from their mid-20s (65 versus 7 µM; $z$-test $P = 1.17 \times 10^{-6}$) and diagnoses codes for E80, K76 (other disorders of liver) and steatohepatitis. Of note, the quantitative phenotypes relevant to the implicated diseases displayed clear recessive patterns (Fig. 4e), confirming the AR nature and the need for homozygous carriers to assess the clinical impact of these variants. These results illustrate G&H as a unique resource to guide variant interpretation for AR disease genes based on the abundance of homozygous genotypes and medical records.

**Insights into drug development from human knockouts.** The presence of human knockouts without major adverse health outcomes suggests that complete lifelong loss of the gene is compatible with viability and that therapeutic antagonism of the gene may likewise be safe and well tolerated[14,16,20]. Using the drug dataset from Open Targets (v23.12)[49], we examined the enrichment of drugs with human knockouts in their target genes over those without per clinical trial phase transition (Methods). Among 3,324 drugs with antagonistic modes of action, drugs with human knockouts were 2.2 times more likely to transition past phase 1 ($P = 5.3 \times 10^{-5}$), the primary focus of which is safety and tolerability (Fig. 4f). This pattern was specific to phase 1 transition, drugs with antagonistic modes of action and drugs with non-oncology indications (Supplementary Fig. 13a and Supplementary Table 20). A comparable association was observed when the analysis was restricted to drugs with a single target gene and when enrichment was examined for drugs with human knockouts in all target genes versus in any target gene (Supplementary Fig. 13b and Supplementary Table 20).

Phenotypic profiles in human knockouts can help inform the therapeutic benefits anticipated by antagonizing a gene. For example, SLC10A2, a bile acid transporter, is targeted by several small-molecule inhibitors primarily to treat biliary diseases. We observed that two individuals homozygous for a frameshift variant (chr13-103052648-CA-C, p.Trp186GlyfsTer23) in *SLC10A2* have markedly reduced low-density lipoprotein (LDL) cholesterol levels (50.3 and 36.3 compared with 106.3 and 116.0 mg dl⁻¹ among matched noncarriers; $z$-test $P = 6.3 \times 10^{-3}$ and $6.2 \times 10^{-6}$, respectively). This is consistent with the significant LDL-cholesterol reduction (13.4–27.0 mg dl⁻¹) observed in the clinical trials of SLC10A2 inhibitors[50-54]. Notably, we observed minimal alteration in the LDL-cholesterol levels among the heterozygous carriers (114.1 compared with 116.0 mg dl⁻¹ among noncarriers, regression $P = 0.93$ adjusting for relevant covariates), suggesting that close to complete loss of SLC10A2 action may be necessary to yield changes in LDL-cholesterol levels. Another example is APOC3, which is targeted by several antisense oligonucleotide drugs to treat hypertriglyceridemia based on its well-understood role in triglyceride metabolism. Consistent with previous reports[19], we found that one individual homozygous for a stop-gain variant (chr11-116830637-C-T, p.Arg19Ter) in *APOC3* had a 62% lower triglyceride level (49.6 compared with 131.1 mg dl⁻¹ among matched noncarriers, $z$-test $P = 0.036$), the magnitude of which is comparable to the range of maximal triglyceride reductions (44–77%) reported in clinical trials of APOC3 inhibition[55-57].

Phenotypic information on human knockouts can also inform the potential safety issues of antagonizing a gene. One example is HSD17B13, which is targeted by several RNA interference therapeutics for the treatment of non-alcoholic liver diseases based on a gene–phenotype association identified through genetic studies[58]. HSD17B13 belongs to the hydroxysteroid (17b) dehydrogenase superfamily involved in steroid metabolism, raising potential risks for reproductive health. We found four (including one from G&H 12k) individuals homozygous for two frameshift variants (chr4-87318347-ATCTCT-A, p.Glu98AspfsTer14 and chr4-87313944-CG-C, p.Ala192AspfsTer14) in *HSD17B13*. Three female carriers had medical records indicating successful pregnancy, with one reporting four healthy pregnancies and

children at a follow-up research visit. This suggests that the absence of HSD17B13 does not severely impact the reproductive potential or pregnancy in females. The remaining male carrier's health record was unremarkable. Overall, this is consistent with the lack of major health or reproductive issues reported in phase 1 trials (NCT04565717 and NCT04202354), which included both male and female participants of reproductive age[59,60]. Another example is IGF1R, which is targeted by several small-molecule inhibitor or antibody drugs for cancer indications. Hyperglycemia was reported as an adverse drug reaction for two inhibitory antibody drugs, while hypoglycemia was reported as a reason for early termination of a trial that tested recombinant IGF1, a ligand of IGF1R (NCT00330668). We found that one individual homozygous for a frameshift variant (chr15-98957361-CGA-C, p.Arg1343ThrfsTer30) in *IGF1R* has markedly higher HbA1c levels (89 compared with 40 mmol mol$^{-1}$ among matched noncarriers, $z$-test $P = 4.3 \times 10^{-4}$, or 47 mmol mol$^{-1}$ among the diabetic subset, $z$-test $P = 0.023$) despite being prescribed three glucose-lowering medications at maximal doses. HbA1c levels were only mildly elevated among the six heterozygous carriers (41 compared with 38 mmol mol$^{-1}$ among noncarriers, regression $P = 0.087$ adjusting for relevant covariates), consistent with the minor increase (1.1 mmol mol$^{-1}$) observed among the heterozygous carriers of rare damaging missense variants in UKB[61].

These results show that the presence of human knockouts and detailed review of their phenotypes can provide meaningful insights for drug development, enabling the assessment of efficacy and possible safety risks of therapeutically targeting a gene.

## Discussion

Our study demonstrates the power of large-scale exome sequencing in a South Asian-ancestry cohort with high autozygosity to drive novel biological discoveries. We identify over 100 previously unreported gene–phenotype associations (with considerable replication in independent datasets) and more than 500 additional genes where homozygous pLoF genotypes are found in adults. We highlight valuable insights that can be gained through the phenotypic review of these human knockouts. As sequencing in G&H scales beyond 100,000 individuals, we anticipate identifying additional genes with human knockouts given the near linear accrual pattern observed thus far[62].

Beyond the rich genomic and medical data currently presented, ongoing initiatives for detailed molecular phenotyping in G&H—encompassing transcriptomics, proteomics and metabolomics—will further enhance the interpretability of genotype–phenotype relationships for these rare genotypes and expand functional allelic series beyond pLoF variants. Another valuable capability in G&H is the ability to recontact individuals for detailed characterization to enable mechanistic insights[14,15]. While profound advances can be made even with a single knockout individual in the presence of strong scientific priors[14-16], recalling first-degree relatives of index knockout individuals can be an effective way to further solidify findings[19,20] (Supplementary Fig. 12d).

Overall, the G&H exome resource presents a valuable opportunity to advance biomedical research within the South Asian community, which has been historically underrepresented in genomic research, while also broadening our understanding of and expanding therapeutic options for human health and disease.

## Online content

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

## Genes and Health Research Team

Klaudia Walter[3,30], Georgios Kalantzis[3,30], Benjamin M. Jacobs[6,30], Joseph Gafton[6,10], Teng Hiang Heng[3], Karen A. Hunt[10], Vivek Iyer[3], Claudia Langenberg[13,14], Daniel G. MacArthur[17,18], Eamonn R. Maher[19], William G. Newman[20,21], Iaroslav Popov[3], Moneeza K. Siddiqui[6], Michael A. Simpson[26], Marie Spreckley[10], John Wright[27], Richard C. Trembath[26], Sarah Finer[6], Hilary C. Martin[3] & David A. van Heel[10,13]

A full list of members and their affiliations appears in the Supplementary Information.

## Methods

### Cohort description

G&H is a longitudinal population genomic medicine study of currently over 65,000 individuals of South Asian ancestry living in the UK, with ongoing recruitment and cohort size target of >100,000 participants[34]. Adult volunteers aged 16 years and over from self-reported British Bangladeshi and British Pakistani ethnicities have been recruited since 2015. At recruitment, each volunteer completes a brief questionnaire, provides an Oragene saliva sample (for DNA and genetic data) and consents for linkage to their longitudinal health records. This includes local primary (general practitioner or family doctor) and secondary (hospital) care electronic health records from UK NHS alongside national datasets from NHS England, which contain Office for National Statistics mortality data (death registry with International Classification of Diseases, 10th Revision (ICD10) coded cause of death), Hospital Episode Statistics data (ICD10 coded inpatient and emergency department diagnoses) and cancer registry data, among others. The demographic characteristics of the cohort have been previously described[34]. The current exome sequencing dataset includes 44,028 individuals, comprising 59% British Bangladeshi, 40% British Pakistani and 1% other South Asian ancestry, with 56% females and 44% males. The work was conducted with approval from the London South East National Research Ethics Service (NRES) Committee of the UK Health Research Authority (14/LO/1240).

### Genetic and phenotypic data preparation

The Supplementary Methods contain details on the generation and quality control of exome sequencing data, calculation of runs of homozygosity, statistical phasing, and the extraction and preparation of phenotypes. Supplementary Tables 20–23 show various details of the variant and genotype quality control.

### Variant annotation

Variants were annotated using the Ensembl Variant Effect Predictor (VEP v105) with LOFTEE plugin (v1.04_GRCh38)[22]. For all analyses, we used the predicted effect on the MANE transcript[21]. pLoF included frameshift, stop-gain, splice acceptor and donor variants and were further annotated by LOFTEE to be high- or low-confidence (pLoF-HC or pLoF-LC, respectively). pDM variants were defined as CADD >20, Polyphen2 >0.445 and SIFT deleterious. There were 138 single variants that had significant associations but did not affect MANE transcripts; we annotated these 138 variants with the most severe effect on an Ensembl canonical transcript (136 variants) or the most severe consequence on a noncanonical transcript (2 variants). We used gnomAD (v4.1) to check for the presence of G&H variants in gnomAD or in the non-Finnish European (NFE) subset of gnomAD, and to compare AF in G&H against that in the gnomAD-NFE subset.

### Clinical variation

We used the ClinVar database[63] (accessed 13 November 13 2022) to annotate disease-relevant variants in G&H exomes. We analyzed variants labeled as pathogenic, likely pathogenic or pathogenic/likely pathogenic in ClinVar (PLP), variants with uncertain significance (VUS), and variants with conflicting interpretations (CI). Disease inheritance patterns were obtained from the OMIM database[48] (accessed 29 April 29 2024). The list of clinically actionable genes was derived from ACMG[64] v3.2. For the comparison of ClinVar annotation against UKB, we downsampled 44,028 European-ancestry exomes from UKB to match the sample size of G&H.

### Fine-scale population structure and founder variants

We defined a set of 17,172 unrelated G&H individuals (9,063 British Bangladeshis and 8,109 British Pakistanis) by removing one individual from each pair if they were related up to third degree and shared IBD segments >40 cM. We used principal component analysis to explore population structure in G&H. We first combined the cohort with a reference panel composed of the 1000 Genomes Project and Pakistanis from the Human Genome Diversity Project (Supplementary Fig. 1a) and then combined it with another reference panel comprising only South Asian individuals from the 1000 Genomes Project and Pakistanis from the Human Genome Diversity Project and BiB (Supplementary Fig. 1b–h). To explore fine-scale structure within the British Pakistanis, we used IBIS[65] to infer IBD regions and then clustered individuals on the basis of their total IBD with other individuals using the Louvain method. We used Fisher's exact tests to identify variants with significantly different frequencies between British Pakistani subpopulations (Supplementary Fig. 2, and Supplementary Tables 1 and 2), considering only the variants that were sufficiently common to see a significant difference after multiple testing given the size of each cluster. Full details are provided in Supplementary Methods.

### Rare variant association analyses under the additive model

We included 54 quantitative traits with measurements in at least 5,000 participants and 591 binary traits with at least 100 cases for rare variant association analyses. We used REGENIE[27] v3 to carry out single-variant and gene-based tests adjusting for age, sex, age$^2$ and the first 20 PCs. For whole genome regression in step 1, we used variants from the exome sequence data with MAF >1%, minor allele count (MAC) >100, missing call rate <10% and Hardy–Weinberg equilibrium $P$ value < $1 \times 10^{-15}$ and samples with missing rate <10%. We also pruned the variants with PLINK using window sizes of 500 variants, shifted by 50 variants, and LD $r^2$ > 0.2. In step 2, we ran a single-variant test for variants with MAC ≥5 and gene-based tests (burden, SKAT and SKAT-O) for four variant consequence masks (Mask A, pLoF-HC; Mask B:, pLoF-HC and pDM; Mask C, all pLoF and missense; Mask D, synonymous) and four AF cutoffs (singletons, <0.01%, <0.1% and <1%). This included 122,690 pLoF-HC variants, 14,958 pLoF-LC variants, 524,289 pDM variants, 1,179,722 other missense variants and 805,502 synonymous variants (SpliceAI <0.1). The number of variants per mask ranged from 66,841 singleton variants in MASK A, up to 1,816,278 variants with AF <0.01 in MASK C (Supplementary Table 21). We applied a Firth correction for associations with $P$ values <0.01.

Genomic inflation factors were calculated as the median of the observed chi-squared test statistics divided by the expected median of the corresponding chi-squared distribution for each test and each phenotype (Supplementary Note 4 and Supplementary Table 7). We used permutation to determine $P$-value cutoffs corresponding to a 5% FDR (Supplementary Methods). Associations driven by synonymous variants and masks, variants in olfactory genes (HGNC 'Olfactory receptors (OR)') or the major histocompatibility complex (MHC, chr6:28510120-33480577), and associations that were not conditionally independent from nearby GWAS associations (Supplementary Methods) were omitted from the reporting of the number of associations.

Because binary traits were derived from ICD10 codes and custom code lists, there was some overlap in phenotype definitions, each with varying sensitivity and specificity. To accurately report the number of unique associations, we merged phenotypes with >75% reciprocal case overlap and applied text matching and manual curation to harmonize trait labels (Supplementary Table 6). All matches were manually reviewed. In total, 675 binary traits were consolidated into 591 unique phenotypes, which were used to report the number of unique associations. Supplementary Tables 7–9 contain results for all 675 binary traits.

We used a custom pipeline to annotate gene-phenotype pairs as 'novel' or 'reported' compared with GWAS/rare variant association studies captured in Open Targets[49] and Genebass[66], and Mendelian diseases reported in OMIM[48] (Supplementary Methods). We assessed replication of significant associations using two datasets: (1) 11,863 additional exomes from G&H (G&H 12k), and (2) ExWAS results using 430,998 European-ancestry exomes of UKB[28] deposited in the GWAS Catalog (Supplementary Methods). For replication, we focused on the

1,191 single-variant and 722 gene-based associations that remained after filtering out associations involving HLA variants/genes, synonymous gene masks, or associations that were not significant after conditioning on nearby GWAS association.

## Meta-analysis with UKB

The analyses in UKB included 425,618 individuals of European-ancestry following the quality control measures previously described[66], including heterozygosity and missing rates, sex concordance and sex chromosome aneuploidy. We conducted rare variant association analyses for 33 cardiometabolic traits, including 13 binary traits and 20 quantitative traits (Supplementary Table 6), adjusting for age, sex, age$^2$, age:sex, age$^2$:sex, exome sequencing batch and top 10 PCs. Comparable approaches to the analyses in G&H were used for whole-genome regression (except for the use of chip genotype data) in step 1 and association testing (including matching mask definitions) in step 2 of REGENIE. Single-variant and gene-based results from UKB and G&H were meta-analyzed with a fixed-effects model weighted by the inverse of standard error using METAL. Genomic control was applied to adjust for population stratification. Results with inconsistent effect directions between UKB and G&H were filtered out.

## Recessive burden analyses

We performed statistical phasing for pLoF and pDM variants with MAF <5% following a previously described approach[43] to identify compound heterozygous genotypes with high confidence (phasing probability >0.9) (Supplementary Methods and Supplementary Fig. 8a). We then aggregated biallelic (homozygous or compound heterozygous) genotypes into recessive burden genotypes per gene (1 if the individual carried a biallelic genotype or 0 otherwise). Recessive gene burdens with at least 4 carriers were tested for association with 54 quantitative phenotypes (with measurements in at least 5,000 individuals) and 439 ICD10 codes (with at least 120 cases) (Supplementary Table 6), adjusting for age, sex, age:sex, age$^2$, age$^2$:sex and top 20 PCs. We performed whole-genome regression in step 1 using the variants filtered by the same parameters as in the additive analyses but in the subset of individuals with corresponding chip genotype data, then tested association with recessive genotype burden in step 2 of REGENIE. We also tested association with additive genotype burden for comparison. We used permutations to derive the $P$-value threshold of significance at $P_{rec} < 2.89 \times 10^{-7}$ corresponding to an FDR of 7.14% (Supplementary Methods and Supplementary Fig. 8b). We additionally considered suggestive associations with $P$ values ($P_{rec} < 5.0 \times 10^{-6}$).

Next, we investigated which of the identified associations exhibited significant dominance deviation using a 2-degree-of-freedom test[13]. Specifically, we examined the following model to jointly test additive and non-additive effects:

$$y = \beta \times g_{add} + \gamma \times g_{domdev} + \text{covariates} + \epsilon,$$

where $g_{add}$ is encoded as [0,1,2] to represent additive genotypes and $g_{domdev}$ is encoded as [0,1,0] to represent dominance genotypes. We used linear or logistic regression for quantitative or binary traits, respectively, using the same covariates among a subset of unrelated individuals. Due to the requirement of using unrelated individuals, the dominance deviation test could be underpowered compared to the mixed-model test by REGENIE which included related individuals. The significance $P$-value threshold was determined by adjusting for the number of associations considered for dominance deviation tests (13 for significant associations and 35 for suggestive associations).

## Identification and analyses of human knockouts

Human knockouts are defined as the homozygous or compound heterozygous carriers of pLoF-HC (MAF <1%) variants within the gene. For the comparison of accrual rate of genes with human knockouts across ancestry groups, we used exome sequence data from UKB[28] with the ancestry label previously reported[67]. The MAF was calculated within each ancestry group to accurately filter out pLoF variants that may exist at a higher frequency in specific ancestry groups. Exome data in each ancestry group were downsampled to match the maximum sample sizes available in different ancestry groups. For the comparison of genes with human knockouts from other genomic datasets, we used the list of genes either reported in previous publications[16,18,19] (note that the criteria used to define human knockouts differ slightly depending on the cohort) or generated on the basis of the variant-level information publicly available in gnomAD[22] (v4.1) using the same criteria used in G&H. The comparison of pLoF genotype distributions between G&H British Pakistanis and Bangladeshis is described in Supplementary Methods. Details of the gene set enrichment analyses are available in Supplementary Methods.

## Phenotypic review of homozygous carriers of pLoF variants in AR disease genes for clinical interpretation

We focused on pLoF variants that are unreported or have unknown significance or conflicting interpretations in ClinVar and that have homozygous carriers in G&H. For statistical testing on quantitative traits, we used $z$-test (in the case of one homozygous individual) or Wilcoxon rank-sum test on log-transformed traits between the homozygous carriers versus noncarrier and heterozygous carriers assuming the recessive effect. We also manually inspected the health record data to examine longitudinal changes in the quantitative traits, diagnostic codes and medication prescriptions relevant to the disease.

## Human knockouts and drug phase transition in clinical trials

Drug target, indication, mode of action and trial phase information was obtained from the Open Targets (v23.12)[49] drug dataset. For each drug, we derived the maximal trial phase and whether any or all of its target genes have human knockouts in G&H. Logistic regression test, along with Fisher's exact test, was used to assess the association between the presence of human knockouts in a drug's target genes and the drug's progression through trial phases, adjusting for the number of target genes. To assess sensitivity, the analyses were repeated stratifying by antagonizing versus agonizing modes of action and oncology versus non-oncology indications, and among the drugs with only one target gene. Drugs listed as inhibitor, antagonist, blocker, negative allosteric modulator, antisense inhibitor, RNAi inhibitor, inverse agonist, disrupting agent, negative modulator, degrader and allosteric antagonist were included as antagonistic drugs, while those listed as activator, agonist, partial agonist, positive allosteric modulator, positive modulator and stabilizer were included as agonistic drugs. Oncology indications were identified by performing a text match for the following terms: 'cancer', 'neoplasia', 'neoplasm', 'leukemia', 'tumor' or words ending in 'oma'.

## Phenotypic review of human knockouts in drug target genes for insights on drug development

To examine the utility of human knockouts in drug development, we focused on drug target genes with antagonistic modes of action that have human knockouts in G&H. The information on indications, adverse drug reactions and safety trial stop reasons was obtained from Open Targets. For statistical analyses, relevant quantitative traits were log transformed, and the lifetime median value in the human knockout was compared with that of an age- (±5 years), sex-, ancestry- (Pakistani or Bangladeshi) and body mass index- (±5 kg m$^{-2}$) matched subset of non-carriers using a two-tailed $z$-test. For statistical analyses between heterozygous carriers and noncarriers, linear regression was used adjusting for age, sex, ancestry and body mass index. Clinical efficacy and safety readouts from the trials were derived from relevant publications.

## Reporting summary

Further information on research design is available in the Nature Portfolio Reporting Summary linked to this article.

# Article

## Data availability

Summary-level data from the G&H 44,028 exomes are publicly available via a Google cloud storage bucket at https://console.cloud.google.com/storage/browser/genesandhealth_publicdatasets/results_44k_ExWAS for web access and gs://genesandhealth_publicdatasets/ for programmatic access. Individual-level data are available only within a Secure Data Environment with controlled access owing to their sensitive nature. Bona fide researchers may obtain access upon application to G&H and approval by the Executive Committee. Detailed instructions can be found at https://www.genesandhealth.org/researchers/apply-for-access/.

## Code availability

Custom codes are available via GitHub (for phasing and identifying biallelic genotypes: https://github.com/BRaVa-genetics/snakemake_pipeline_for_phasing, for recessive association and related analyses: https://github.com/giorkala/gnh_flagship_recessive, TREtools for trait extraction and preparation: https://github.com/genes-and-health/tre-tools) and via Zenodo at https://doi.org/10.5281/zenodo.18445301 (ref. 68).

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

## Acknowledgements

G&H is/has recently been core-funded by Wellcome (WT102627, WT210561), the Medical Research Council (UK) (M009017, MR/X009777/1, MR/X009920/1), Higher Education Funding Council for England Catalyst, Barts Charity (845/1796), Health Data Research UK (for London substantive site) and research delivery support from the NHS National Institute for Health Research Clinical Research Network (North Thames). We acknowledge the support of the National Institute for Health and Care Research Barts Biomedical Research Centre (NIHR203330); a delivery partnership of Barts Health NHS Trust, Queen Mary University of London, St George's University Hospitals NHS Foundation Trust and St George's University of London. Current members of the Genes & Health Research Team are listed in full in Supplementary Note 9. Exome sequencing and analyses are funded by the Genes and Health Industry Consortium 1 of AstraZeneca PLC, Bristol-Myers Squibb Company, GlaxoSmithKline Research and Development Limited, Maze Therapeutics Inc., Merck Sharp & Dohme LLC, a subsidiary of Merck & Co., Inc., Rahway, NJ, USA, Novo Nordisk A/S, Pfizer Inc., and Takeda Development Centre Americas Inc. G&H is/has also recently been funded by Alnylam Pharmaceuticals, 5 Prime Life Sciences, Phaeron, and Genomics PLC. H.C.M., K.W. and G.K. are partly supported by the funding from Wellcome Trust (220540/Z/20/A) and Wellcome Sanger Institute Quinquennial Review 2021–2026. S.O. acknowledges funding from the Wellcome Trust (226800/Z/22/Z and 314234/Z/24/Z) and NIHR Cambridge Biomedical Research Centre NIHR203312. F.H.L. is supported by the Wellcome Trust award (224894/Z/21/Z) and the Medical Sciences Doctoral Training Centre at the University of Oxford. D.S.P. is supported by the Wellcome Trust Investigator award (221782/Z/20/Z). We thank Social Action for Health, Centre of The Cell, members of our Community Advisory Group, and staff who have recruited and collected data from volunteers. We thank the NIHR National Biosample Centre (UK Biocentre), the Social Genetic & Developmental Psychiatry Centre (King's College London), Wellcome Sanger Institute and Broad Institute for sample processing, genotyping, sequencing and variant annotation. This work uses data provided by patients and collected by the NHS as part of their care and support. This research utilized Queen Mary University of London's Apocrita HPC facility, supported by QMUL Research-IT (https://doi.org/10.5281/zenodo.438045). We thank the Barts Health NHS Trust, NHS Clinical Commissioning Groups (City and Hackney, Waltham Forest, Tower Hamlets, Newham, Redbridge, Havering, Barking and Dagenham), East London NHS Foundation Trust, Bradford Teaching Hospitals NHS Foundation Trust, Public Health England (especially D. Wyllie), Discovery Data Service/Endeavour Health Charitable Trust (especially D. Stables), Voror Health Technologies Ltd (especially S. Don) and NHS England (for what was NHS Digital)—for General Data Protection Regulation-compliant data sharing backed by individual written informed consent. Most of all, we thank all the volunteers participating in G&H. A favorable ethical opinion for the main G&H research study was granted by NRES Committee London – South East (reference 14/LO/1240) on 16 September 2014. Queen Mary University of London is the Sponsor and Data Controller. Analyses with UKB data were conducted under application IDs 26041 and 50314. For the purpose of open access, the authors have applied a CC-BY public copyright license to any author-accepted manuscript version arising from this submission.

## Author contributions

Conception and scientific oversight: H.I.K., H.C.M. and D.A.v.H. Exome sequence data generation and quality control: D.A.v.H., I.P., T.H.H., N.G. and M.A.S. Phenotype extraction and quality control: B.M.J., D.A.v.H., J.M.D. and K.E. Variant and population genetic analyses: K.W., H.C.M., D.A.v.H., K.K. and A.Y.D. Rare variant association analyses and downstream analyses: C.D., K.W., D.A.v.H., P.M.-S., A.M.M., K.A.C., G.K. and B.G. Meta-analysis and downstream analyses: C. Li, S.V.M. and C.D. Recessive burden analyses: G.K., F.H.L., D.S.P. and H.C.M. Human knockout identification and downstream analyses: H.I.K., G.K. and M.A.A. Medical record review: B.M.J., J.G., S.F. and D.A.v.H. Clinical variant analyses: K.K., H.I.K., B.M.J. and G.d.A. Intellectual contribution: C. Langenberg, M.K.S., D.G.M., S.O., R.R.G. and C.M. Consortium leadership and supervision: D.A.v.H., S.F., R.C.T., G.d.A., S.P., E.R.H., J.C.M., L.A., R.M.T., K.E., S.L., J.M.M.H., Y.J., E.B.F., H.I.K., M.R.M. and D.D. Cohort operations: K.A.H., D.A.v.H., S.F., W.G.N., E.R.M. and J.W. Consortium project management: M.S. Consortium informatics: V.I. Writing—original draft: H.I.K., C.D., K.W., G.K., C. Li, S.V.M., K.K., B.M.J., H.C.M. and D.A.v.H. Writing—review and editing: all authors.

## Competing interests

H.I.K., M.A.A., E.B.F. and M.R.M. are employees and/or stockholders of Pfizer. C.D., S.V.M., R.R.G. and K.E. are employees and/or stockholders of Maze Therapeutics. C. Li, K.A.C., E.R.H. and J.C.M. are employees and/or stockholders of Bristol-Myers Squibb. K.K., G.d.A. and S.P. are employees and/or stockholders of AstraZeneca. P.M.-S. and D.D. are employees and/or stockholders of Takeda. A.M.M., B.G. and S.L. are employees of Merck Sharp & Dohme LLC, a subsidiary of Merck & Co., Inc., Rahway, NJ, USA, and/or stockholders of Merck & Co., Inc., Rahway, NJ, USA. J.M.D., L.A. and R.M.T. are employees and/or stockholders of GSK. A.Y.D., C.M., Y.J. and J.M.M.H. are employees and/

or stockholders of Novo Nordisk. W.G.N. is an employee of Fava Health. The other authors declare no competing interests.

## Additional information

**Extended data** is available for this paper at https://doi.org/10.1038/s41588-026-02553-7.

**Correspondence and requests for materials** should be addressed to Hye In Kim, Hilary C. Martin or David A. van Heel.

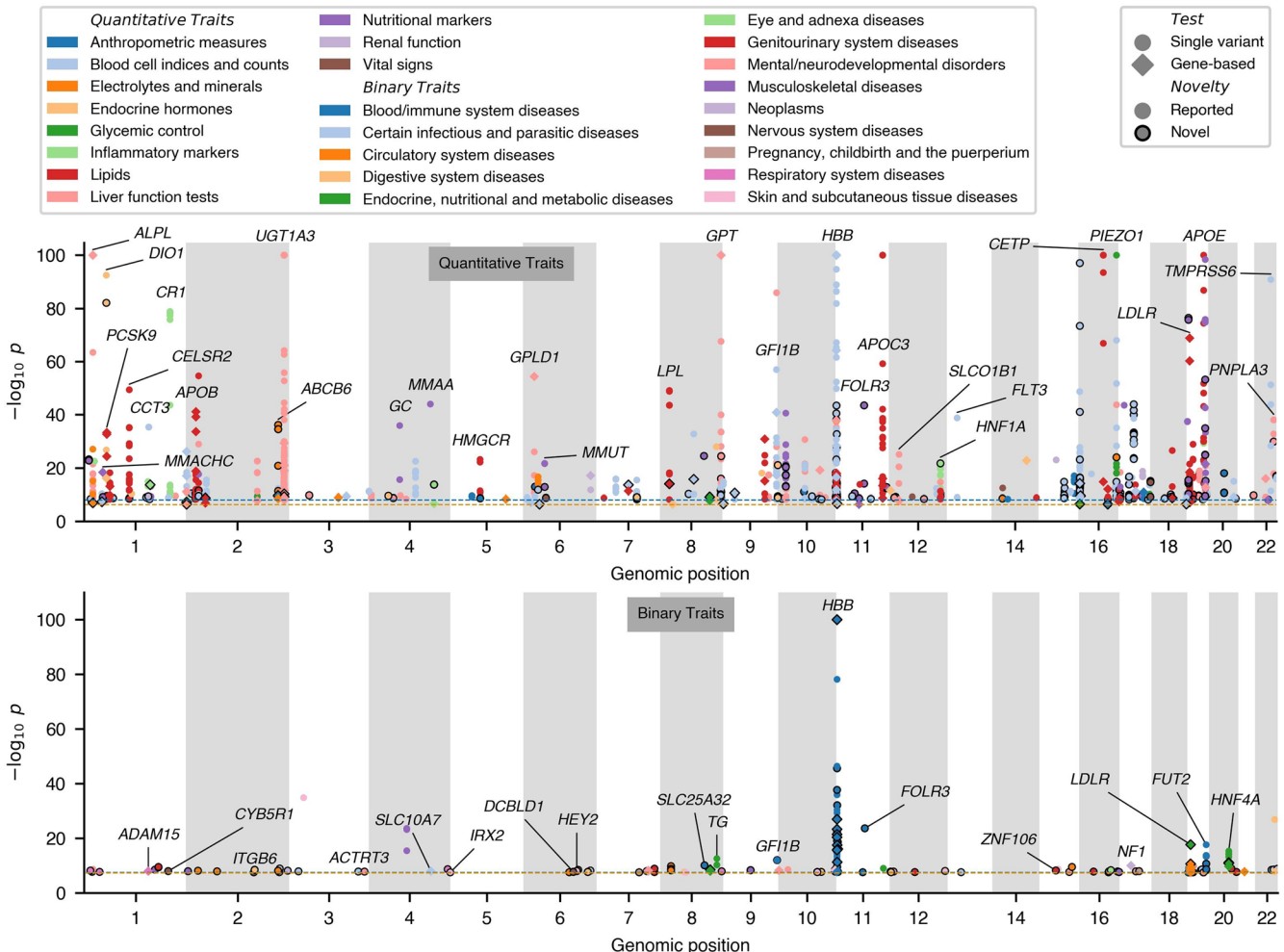

**Extended Data Fig. 1 | Significant associations from exome-wide association analyses of 44,028 G&H exomes.** Significant associations for quantitative (top) and binary (bottom) traits in Manhattan plots. The *y*-axis indicates −log$_{10}$ association *P*-values and is truncated at 100. Fill color indicates phenotype categories. Marker shape indicates single variant (circle) and gene-based (diamond) associations. Marker outline indicates gene-phenotype associations annotated as novel. Blue dashed lines indicate single variant *P*-value cutoffs. Orange dashed lines indicate gene-based *P*-value cutoffs.

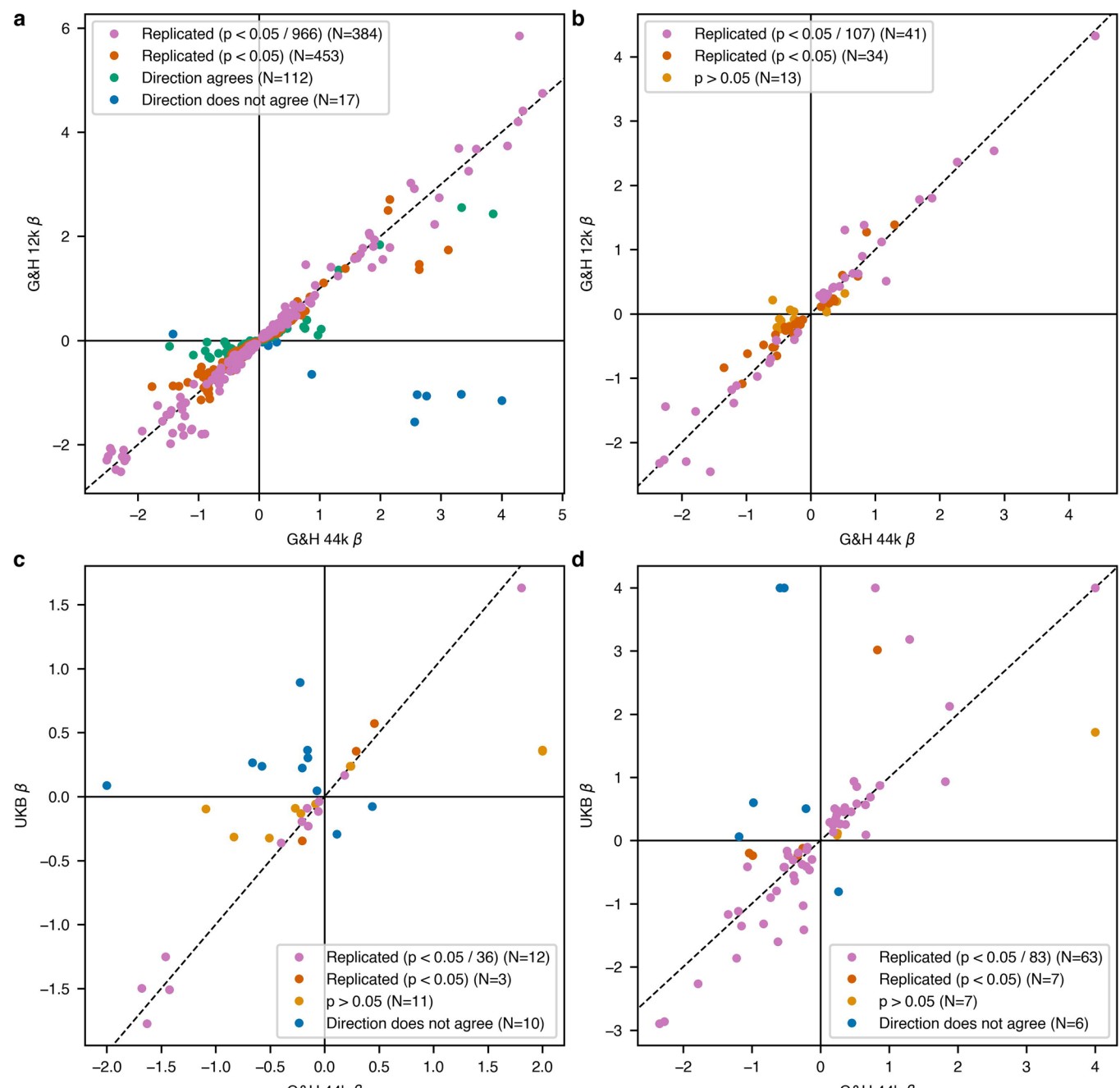

**Extended Data Fig. 2 | Replication of significant associations from G&H ExWAS. a,b,** Comparison of effect sizes between G&H 44k discovery and G&H 12k replication sets for the significant single variant (**a**) and gene-based (**b**) associations from G&H 44k discovery. **c,d,** Comparison of effect sizes between G&H 44k discovery and UK Biobank replication sets for the significant single variant (**c**) and gene-based (**d**) associations from G&H 44k discovery. Marker color indicates the replication outcome based on the association *P*-values from the replication set (Bonferroni corrected or nominal) and the concordance of effect directions between the discovery and replication sets. Absolute effect sizes are truncated at 2 in **c** and 4 in **d**. Note, gene-based results that do not have effect sizes (SKAT, SKAT-O) are not plotted.

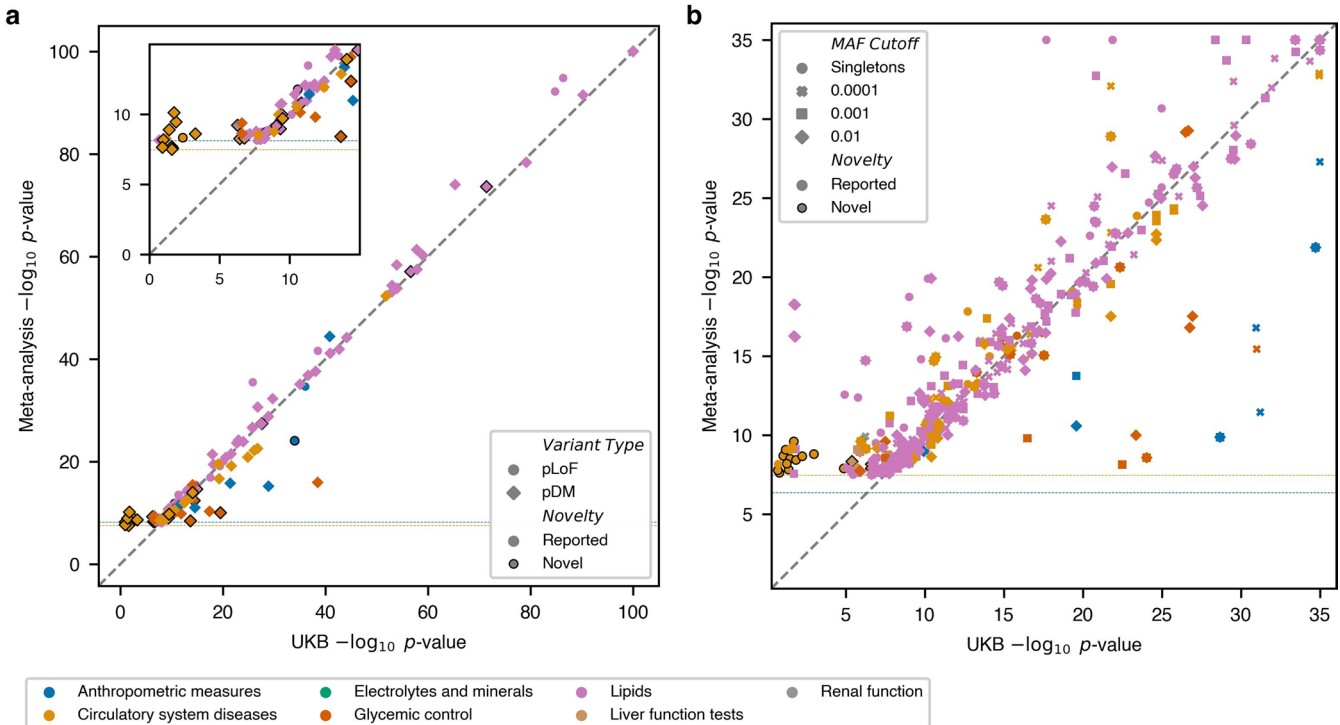

**Extended Data Fig. 3 | Contribution of G&H to the statistical power gain by meta-analysis. a**, Comparison of −log₁₀ association *P*-values between the meta-analysis and the UKB-only analysis for the significant single variant pLoF/pDM associations from meta-analysis. −log₁₀*P*-values were truncated at 100. Marker shape indicates variant type. **b**, Comparison of −log₁₀ association *P*-values between the meta-analysis and the UKB-only analysis for the significant

gene-based associations from meta-analysis. −log₁₀*P*-values were truncated at 35. Marker shape indicates variant frequency cutoff used for the mask. Marker color indicates phenotype categories. Dark marker outline indicates that the gene-phenotype association is annotated as novel. Blue and orange dashed lines indicate QT and BT *P*-value cutoffs, respectively.

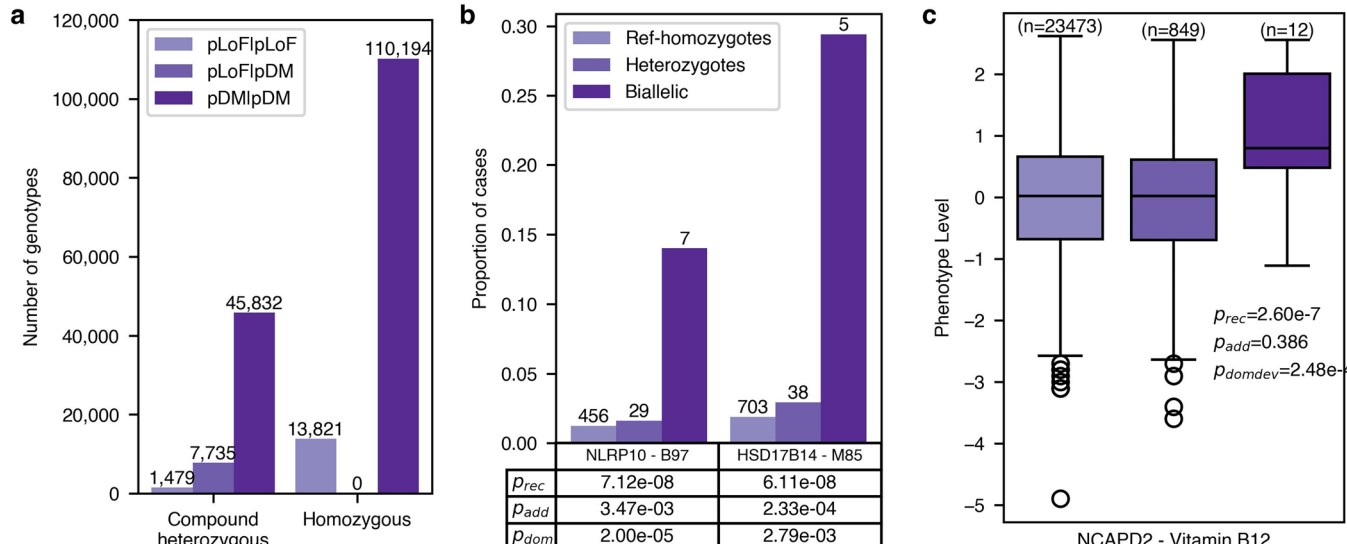

**Extended Data Fig. 4 | Number of biallelic genotypes and significant associations from recessive burden analyses. a**, The number of compound heterozygous and homozygous genotypes found among the 39,148 phased exomes stratified by variant consequence. **b**, Bar plots showing the proportion of diseased individuals per pLoF/pDM burden genotype for the significant recessive associations with binary traits. The number of diseased carriers is shown at the top of each bar and the association P-values from recessive ($P_{rec}$), additive ($P_{add}$), and dominance deviation ($P_{domdev}$) tests are shown in the inset table. **c**, Box plots showing the quantile normalized levels of covariate adjusted phenotypes per pLoF/pDM burden genotype for a significant recessive association with a quantitative trait. The center line indicates the median. The edges of the box indicate the first and third quartiles. The whiskers extend to the most extreme data points within 1.5 times the interquartile range. The points indicate the datapoints that fall outside the range of the whiskers. The number of carriers is indicated at the top of each box, and the association P-values from recessive ($P_{rec}$), additive ($P_{add}$), and dominance deviation ($P_{domdev}$) tests are shown within the plot.

# Reporting Summary

## Statistics

For all statistical analyses, confirm that the following items are present in the figure legend, table legend, main text, or Methods section.

| n/a | Confirmed | |
|---|---|---|
| ☐ | ☒ | The exact sample size (*n*) for each experimental group/condition, given as a discrete number and unit of measurement |
| ☐ | ☒ | A statement on whether measurements were taken from distinct samples or whether the same sample was measured repeatedly |
| ☐ | ☒ | The statistical test(s) used AND whether they are one- or two-sided<br>*Only common tests should be described solely by name; describe more complex techniques in the Methods section.* |
| ☐ | ☒ | A description of all covariates tested |
| ☐ | ☒ | A description of any assumptions or corrections, such as tests of normality and adjustment for multiple comparisons |
| ☐ | ☒ | A full description of the statistical parameters including central tendency (e.g. means) or other basic estimates (e.g. regression coefficient) AND variation (e.g. standard deviation) or associated estimates of uncertainty (e.g. confidence intervals) |
| ☐ | ☒ | For null hypothesis testing, the test statistic (e.g. $F$, $t$, $r$) with confidence intervals, effect sizes, degrees of freedom and $P$ value noted<br>*Give P values as exact values whenever suitable.* |
| ☒ | ☐ | For Bayesian analysis, information on the choice of priors and Markov chain Monte Carlo settings |
| ☒ | ☐ | For hierarchical and complex designs, identification of the appropriate level for tests and full reporting of outcomes |
| ☐ | ☒ | Estimates of effect sizes (e.g. Cohen's *d*, Pearson's *r*), indicating how they were calculated |

*Our web collection on statistics for biologists contains articles on many of the points above.*

## Software and code

Policy information about availability of computer code

| Data collection | gnomAD (v4.1)<br>ClinVar (accessed November 13th, 2022)<br>OMIM (accessed April 29th, 2024)<br>ACMG (v3.2)<br>Open Targets (v23.12)<br>Genebass (static) |
|---|---|
| Data analysis | Variant calling pipeline: https://broadinstitute.github.io/warp/docs/Pipelines/Exome_Germline_Single_Sample_Pipeline/<br>Joint genotyping pipeline: https://broadinstitute.github.io/warp/docs/Pipelines/JointGenotyping_Pipeline/<br>Ensembl Variant Effect Prediction (v105) with LOFTEE plugin (v1.04_GRCh38):<br>PLINK (v2.0) for basic operations on genotype data: https://www.cog-genomics.org/plink/2.0/<br>REGENIE (v3) for association testing: https://rgcgithub.github.io/regenie/<br>TREtools for trait extraction and preparation: https://github.com/genes-and-health/tre-tools<br>Custom codes for phasing and identifying biallelic genotypes: https://github.com/BRaVa-genetics/snakemake_pipeline_for_phasing<br>Custom codes for recessive association and other analyses: https://github.com/giorkala/gnh_flagship_recessive |

For manuscripts utilizing custom algorithms or software that are central to the research but not yet described in published literature, software must be made available to editors and reviewers. We strongly encourage code deposition in a community repository (e.g. GitHub). See the Nature Portfolio guidelines for submitting code & software for further information.

## Data

Policy information about availability of data

All manuscripts must include a data availability statement. This statement should provide the following information, where applicable:
- Accession codes, unique identifiers, or web links for publicly available datasets
- A description of any restrictions on data availability
- For clinical datasets or third party data, please ensure that the statement adheres to our policy

Summary-level data from the G&H 44,028 exomes are publicly available in a Google cloud storage bucket: https://console.cloud.google.com/storage/browser/genesandhealth_publicdatasets/results_44k_ExWAS for web access and gs://genesandhealth_publicdatasets/ for programmatic access. Individual-level data are only available within a Secure Data Environment with controlled access due to their sensitive nature. Bona fide researchers may obtain access upon application to G&H and approval by the Executive Committee. Detailed instructions can be found at https://www.genesandhealth.org/researchers/apply-for-access/.

## Research involving human participants, their data, or biological material

Policy information about studies with human participants or human data. See also policy information about sex, gender (identity/presentation), and sexual orientation and race, ethnicity and racism.

| Reporting on sex and gender | Provided in the manuscript. |
|---|---|
| Reporting on race, ethnicity, or other socially relevant groupings | Provided in the manuscript. |
| Population characteristics | Adult British volunteers of self-reported South Asian ancestry. The median age of recruitment was 39 and 56% were females among the 44,028 participants analyzed in the study. More details can be found in the cohort profile published previously (https://doi.org/10.1093/ije/dyz174). |
| Recruitment | British Bangladeshi and Pakistani individuals aged 16 and over are invited for voluntary participation. Recruitment largely took place in community settings or health care settings. More details can be found in a prior cohort profile paper (https://doi.org/10.1093/ije/dyz174). Self-selection, convenience sampling, or health status biases may limit the generalizability or the transferability of the findings. |
| Ethics oversight | The study was approved by the London South East NRES Committee of the UK Health Research Authority (14/LO/1240). |

Note that full information on the approval of the study protocol must also be provided in the manuscript.

# Field-specific reporting

Please select the one below that is the best fit for your research. If you are not sure, read the appropriate sections before making your selection.

☒ Life sciences    ☐ Behavioural & social sciences    ☐ Ecological, evolutionary & environmental sciences

For a reference copy of the document with all sections, see nature.com/documents/nr-reporting-summary-flat.pdf

# Life sciences study design

All studies must disclose on these points even when the disclosure is negative.

| Sample size | Up to 44,028 pending on the availability of the relevant phenotype data (not determined a priori). |
|---|---|
| Data exclusions | No exclusions except for predefined standard QC metrics as described in the Methods and the Supplementary Methods. |
| Replication | All significant associations found in this study were examined for replication in two independent datasets as described in the manuscript. |
| Randomization | Not relevant as this is an observational study. |
| Blinding | Not relevant as this is an observational study. |

# Reporting for specific materials, systems and methods

We require information from authors about some types of materials, experimental systems and methods used in many studies. Here, indicate whether each material, system or method listed is relevant to your study. If you are not sure if a list item applies to your research, read the appropriate section before selecting a response.

## Materials & experimental systems

| n/a | Involved in the study |
|-----|----------------------|
| ☒ ☐ | Antibodies |
| ☒ ☐ | Eukaryotic cell lines |
| ☒ ☐ | Palaeontology and archaeology |
| ☒ ☐ | Animals and other organisms |
| ☒ ☐ | Clinical data |
| ☒ ☐ | Dual use research of concern |
| ☒ ☐ | Plants |

## Methods

| n/a | Involved in the study |
|-----|----------------------|
| ☒ ☐ | ChIP-seq |
| ☒ ☐ | Flow cytometry |
| ☒ ☐ | MRI-based neuroimaging |

## Plants

| | |
|---|---|
| Seed stocks | *Report on the source of all seed stocks or other plant material used. If applicable, state the seed stock centre and catalogue number. If plant specimens were collected from the field, describe the collection location, date and sampling procedures.* |
| Novel plant genotypes | *Describe the methods by which all novel plant genotypes were produced. This includes those generated by transgenic approaches, gene editing, chemical/radiation-based mutagenesis and hybridization. For transgenic lines, describe the transformation method, the number of independent lines analyzed and the generation upon which experiments were performed. For gene-edited lines, describe the editor used, the endogenous sequence targeted for editing, the targeting guide RNA sequence (if applicable) and how the editor was applied.* |
| Authentication | *Describe any authentication procedures for each seed stock used or novel genotype generated. Describe any experiments used to assess the effect of a mutation and, where applicable, how potential secondary effects (e.g. second site T-DNA insertions, mosiacism, off-target gene editing) were examined.* |

