## [Peer Review File · Nature Genetics]

Exome sequencing and analysis of 44,028 British South Asians enriched for high autozygosity

Corresponding Author: Dr Hye In Kim

Version 0:

Decision Letter:

30th July 2025

Dear Hye In,

Your Article "Exome sequencing and analysis of 44,028 British South Asians enriched for high autozygosity" has been seen by two referees. You will see from their comments below that, while they find your work of interest, they have raised several relevant points. We are interested in the possibility of publishing your study in Nature Genetics, but we would like to consider your response to these points in the form of a revised manuscript before we make a final decision on publication.

To guide the scope of the revisions, the editors discuss the referee reports in detail within the team, including with the chief editor, with a view to identifying key priorities that should be addressed in revision, and sometimes overruling referee requests that are deemed beyond the scope of the current study. In this case, we ask that you address all technical points related to the genetic analyses by contextualizing the findings, adding replication testing where feasible, revising claims where needed, and clarifying presentation throughout. We hope you will find this prioritized set of referee points to be useful when revising your study. Please do not hesitate to get in touch if you would like to discuss these issues further.

We therefore invite you to revise your manuscript taking into account all reviewer and editor comments. Please highlight all changes in the manuscript text file. At this stage, we will need you to upload a copy of the manuscript in MS Word .docx or similar editable format.

*2) If you have not done so already, please begin to revise your manuscript so that it conforms to our Article format instructions, available

[here](http://www.nature.com/ng/authors/article_types/index.html).

*3) Include a revised version of your Reporting Summary: <https://www.nature.com/documents/nr-reporting-summary.pdf>

It will be available to referees (and, potentially, statisticians) to aid in their evaluation if the manuscript goes back for peer review. A revised Reporting Summary is essential for re-review of the paper.

Please be aware of our [guidelines](https://www.nature.com/nature-research/editorial-policies/image-integrity) on digital image standards.

EXTENDED DATA FIGURES

When re-submitting your manuscript, please ensure that any supplementary figures and tables that are crucial to the manuscript's conclusions are converted into Extended Data figures and tables to increase visibility of these data. Extended

Data figures and tables are online-only (present in the online PDF and full-text HTML versions of the paper), peer-reviewed display items that provide essential background to the article but are not included in the main article due to space constraints. A maximum of ten Extended Data display items (figures and tables) is permitted.

Link Redacted

We hope to receive your revised manuscript within 8-12 weeks. If you cannot send it within this time, please let us know.

Nature Genetics is committed to improving transparency in authorship. As part of our efforts in this direction, we are now requesting that all authors identified as 'corresponding author' on published papers create and link their Open Researcher and Contributor Identifier (ORCID) with their account on the Manuscript Tracking System (MTS), prior to acceptance. ORCID helps the scientific community achieve unambiguous attribution of all scholarly contributions. You can create and link your ORCID from the home page of the MTS by clicking on 'Modify my Springer Nature account'. For more information, please visit www.springernature.com/orcid.

Sincerely,
Kyle

Kyle Vogan, PhD
Senior Editor
Nature Genetics
<https://orcid.org/0000-0001-9565-9665>

Referee expertise:

Referee #1: Genetics, complex traits, underrepresented populations

Referee #2: Genetics, complex traits, population-based sequencing

Reviewers' Comments:

Reviewer #1 (Remarks to the Author):

This is an in-depth body of work by Kim et al. on an underrepresented population including exome sequencing on ~44K British-Pakistani and British-Bangladeshi individuals tied to their medical records. Overall, I found the work to be comprehensive and robust. I have two major comments and several minor ones that mostly have to do with areas needing clarification.

My first major feedback is that it is important for these results (in all places) to have differences noted in the G&H group more clearly laid out with respect to genetic ancestry vs. autozygosity. Firstly, the authors reference work documenting high autozygosity in these populations, but description of the noted levels in the data at hand is important for context. Even the title references 'high autozygosity', but what is 'high'? What impact does autozygosity vs. ancestry play in the overall findings? Some examples of places there are statements to this effect but the evidence is not really provided are:

(1) "High autozygosity in G&H can provide greater statistical power for recessive association analyses which have been relatively less explored. There were 13,821 and 110,194 homozygous genotypes for pLoF and pDM variants with MAF<5%, respectively (Supplementary Table 13)." Can you contextualize this relative to other populations where there is no elevated autozygosity?

(2) Figure 4 panel A suggests that all groups are different from EUR, so please redo to perhaps truncate the X axis. Also clarify is this all or only independents in G&H? Once zoomed in, we need to understand if G&H is indeed different from EUR AND different from the other populations, i.e., is this 'excess' a function of autozygosity?

My second major point is that I appreciate the depth of analysis. There are a lot of results, and the authors have done well in making all available with driving home key messages using some illustrative results. That said, there is a lot packed into each main figure, and this is at the expense of resolution and clarity. Figure 2 panels A and B are a real challenge in their

current format. Similarly Figure 3 panel C. Perhaps the 'main message' is what these should focus on without the attempt to present all break-downs in a single figure that is part of a multi-panel display.

"For the comparison of ClinVar annotation against UK Biobank, we down-sampled the European-ancestry exomes from UK Biobank to match the sample size of G&H." Which G&H - total or independents?

"British-" is sometimes hyphenated and sometimes not.

"Overall, 267 of the 3,096 significant single-variant associations involved variants that were significantly enriched in specific British Pakistani sub-populations, and 82% of these associations were for HLA variants (Supplementary Note 1)." Please check and rewrite. There is a typo for the word "that" leading to an incomplete thought, and what does the 82% reference? Additionally, please clarify counts of associations to stand alone in text. This sentence says 3,096 in all but then 1,139 and 159 for QT and BT in below paragraphs? This may be in part due to HLA included/not excluded, but this could be simplified/clarified.

"Associations for these circulatory system diseases had stronger p-values in G&H compared to UKB, despite the nearly 10-fold difference in sample size between the two cohorts (Supplementary Figure 7)". This is hard to capture given the range of the axes. I get the point of setting X=Y on axis, but if the x=y line is drawn so perhaps scale Y axis to see that the higher p-value occurrences are largely on the Circ traits. This is better illustrated for example in panel 3C insert?

"We used principal component analysis to explore population structure in G&H, combining the cohort with a reference panel composed of South Asian individuals from other cohorts (Supplementary Figure 1)." This needs to be reconciled as the actual Figure S1 Panel A in SF1 is inclusive of all 1000GP so does not align with the statement of methods on reference panel used.

Please provide more detail on the variants including in the rare variant analysis in UKBB especially as a variety of masks were used in G&H for gene based tests. So were the same options used in UKBB?

A couple of unclear results summarized in text and Figure 3: the marker size must reflect some range not the exact p-value? If the thresholds are 'less than' then only variants with $p < 0.1$ are shown. What about variants that were not < 0.1 but that went into the gene-based tests?

Reviewer #2 (Remarks to the Author):

Review of the manuscript "Exome sequencing and analysis of 44,028 British South Asians enriched for high autozygosity" by Kim et al. (Nature Genetics submission)

The manuscript focused on a population where consanguinity is high, giving a unique opportunity to assess first the effect of "human knockouts", otherwise vanishingly rare in other population.

Overall, the study is of great interest, but the results are sometimes presented as significant based mainly on a single study group. The authors would benefit from moving some of the least significant result to supplementary note. In several instances, the author claimed to be the first to find something, which I find unusual and unnecessary, especially when it appears that this claim is wrong.

Specific comments

1/ When mentioning actionable genotypes in ACMG 81 genes, authors refer to PLP (pathogenic and likely pathogenic variants). Do they mean reported as P or LP or do they classify them according to guidelines (including novel LOF in genes where LOF are pathogenic)?

2/ The authors use permutation to determine significance thresholds, which is somewhat uncommon in the field. Can they present what the threshold would be just correcting for multiple testing (Bonferroni)?

3/ Concerning the gene burden test models, it is not clear how many in total are tested (combo of model type and frequency). Often these tests are correlated and P value Cauchy have been proposed to correct for multiple testing.

4/ In the following points, replication is needed. When testing burden test in about 40K here, why was the result not also present in UKB where close to 10 times more individuals were sequenced? The value of this study is particularly the homozygotes, i.e. human knockouts, the heterozygotes are better studied in other populations.

5/ For ADAM15, how many carriers are counted for the burden tests and how many are affected? Was it also tested in the UKB? Did it replicate there? The fact that most singleton are population specific is not surprising and is a consequence of their rare nature.

6/ MMACHC: the authors claim that "this is the first association of coding variants with vitamin B12 levels". Unfortunately it is not correct. Association of R206Q in MMACHC with B12 was reported over a decade ago in <https://journals.plos.org/plosgenetics/article?id=10.1371/journal.pgen.1003530>. In that publication: "the rare missense rs12272669 variant (MAF 0.22%) in MMACHC that associates with B12 found in the Icelandic data..." Interesting the normalized effects are very similar (about half an SD).

7/ LMNA: association of rare variants to atrial fibrillation is interesting but somewhat not surprising given that rare variants in LMNA are known to cause rare cardiac diseases. Is it possible that here AF diagnosis is a potential differential diagnosis?

8/ When discussing homozygotes in the chapter under "Insights from human knockouts", most of the claims are not supported statistically and are based on a single individual. The authors tend, to my opinion, to overinterpret small differences of 1 individual to the median. I would move most of these claims to supplementary notes or support them statistically.

9/ Are they LOF variants that have a frequency high enough that in this population with high consanguinity we would expect to detect some homozygotes whereas none are observed?

10/ Absence of Replication in general and strict significance are needed to make claims.

Version 1:

Decision Letter:

Our ref: NG-A69308R

20th December 2025

Dear Hye In,

Your revised manuscript "Exome sequencing and analysis of 44,028 British South Asians enriched for high autozygosity" (NG-A69308R) has been seen by the original referees. As you will see from their comments below, they find that the paper has improved in revision, and therefore we will be happy in principle to publish it in Nature Genetics as an Article pending final revisions to satisfy Reviewer #2's remaining request and to comply with our editorial and formatting guidelines.

We are now performing detailed checks on your paper, and we will send you a checklist detailing our editorial and formatting requirements soon. Please do not upload the final materials or make any revisions until you receive this additional information from us.

Thank you again for your interest in Nature Genetics. Please do not hesitate to contact me if you have any questions.

Sincerely,
Kyle

Kyle Vogan, PhD
Senior Editor
Nature Genetics
<https://orcid.org/0000-0001-9565-9665>

Reviewer #1 (Remarks to the Author):

The authors have addressed all prior concerns and I have no additional comments.

Reviewer #2 (Remarks to the Author):

The authors addressed the reviewer's comments in a satisfactory manner.

Concerning the multiple testing threshold through Bonferroni correction, even if very conservative, please mention how many findings would reach this threshold.
